# SoundnessBench: A Soundness Benchmark for Neural Network Verifiers

**Xingjian Zhou**[1]*  
*University of California, Los Angeles*  
*hzhou27@g.ucla.edu*

**Keyi Shen**[2]*  
*Georgia Institute of Technology*  
*kshen84@gatech.edu*

**Andy Xu**[1]*  
*University of California, Los Angeles*  
*xuandy05@gmail.com*

**Hongji Xu**[3]*  
*Duke University*  
*hx84@duke.edu*

**Cho-Jui Hsieh**[1]  
*University of California, Los Angeles*  
*chohsieh@cs.ucla.edu*

**Huan Zhang**[4]  
*University of Illinois Urbana-Champaign*  
*huan@huan-zhang.com*

**Zhouxing Shi**[5]  
*University of California, Riverside*  
*zhouxing.shi@ucr.edu*

**Reviewed on OpenReview:** *https://openreview.net/forum?id=UuYYldVLH3*

## Abstract

Neural network (NN) verification aims to formally verify properties of NNs, which is crucial for ensuring the behavior of NN-based models in safety-critical applications. In recent years, the community has developed many NN verifiers and benchmarks to evaluate them. However, existing benchmarks typically lack ground-truth for hard instances where no current verifier can verify the property and no counterexample can be found. This makes it difficult to validate the soundness of a verifier, when it claims verification on such challenging instances that no other verifier can handle. In this work, we develop a new benchmark for NN verification, named "**SoundnessBench**", specifically for testing the soundness of NN verifiers. SoundnessBench consists of instances with deliberately inserted counterexamples that are hidden from adversarial attacks commonly used to find counterexamples. Thereby, it can identify false verification claims when hidden counterexamples are known to exist. We design a training method to produce NNs with hidden counterexamples and systematically construct our SoundnessBench with instances across various model architectures, activation functions, and input data. We demonstrate that our training effectively produces hidden counterexamples and our SoundnessBench successfully identifies bugs in state-of-the-art NN verifiers. Our code is available at `https://github.com/mvp-harry/SoundnessBench` and our dataset is available at `https://huggingface.co/datasets/SoundnessBench/SoundnessBench`.

---

*Equal contribution

# 1 Introduction

As neural networks (NNs) have demonstrated remarkable capabilities in various applications, formal verification for NNs has attracted much attention, due to its importance for the trustworthy deployment of NNs with formal guarantees, particularly in safety-critical applications. Typically the goal of NN verification is to formally check if the output by an NN provably satisfies particular properties, for any input within a certain range. In recent years, many softwares specialized for formally verifying NNs, known as NN verifiers, have been developed, such as $\alpha,\beta$-CROWN (Zhang et al., 2018; Xu et al., 2020; 2021; Wang et al., 2021; Zhang et al., 2022; Shi et al., 2025), Marabou (Katz et al., 2019; Wu et al., 2024), NeuralSAT (Duong et al., 2024), VeriNet (Henriksen & Lomuscio, 2020), nnenum (Bak, 2021), NNV (Tran et al., 2020), MN-BaB (Ferrari et al., 2022), etc.

To evaluate the performance of various NN verifiers and incentivize new development, many benchmarks have been proposed. Additionally, there is a series of standardized competitions, International Verification of Neural Networks Competition (VNN-COMP) (Bak et al., 2021; Müller et al., 2022; Brix et al., 2023a;b; 2024), that compile diverse benchmarks to evaluate NN verifiers. However, existing benchmarks have a crucial limitation: their instances lack "ground-truth" regarding whether the property in an instance can be verified or falsified. This makes it difficult to determine whether a verification claim is sound when no existing verifier can falsify the properties being checked. For example, in VNN-COMP 2023 (Brix et al., 2023a), a benchmark of Vision Transformers (ViT) (Dosovitskiy et al., 2021) proposed by a participating $\alpha,\beta$-CROWN team (Shi et al., 2025) did not contain any instance falsifiable by the participants, while another participating Marabou (Wu et al., 2024) team showed 100% verification on this benchmark, which was later found to be unsound due to issues when leveraging an external solver (Brix et al., 2023a). This limitation of existing benchmarks also makes it difficult for developers of NN verifiers to promptly identify bugs, due to the error-prone nature of NN verification.

To address this limitation, we propose a novel approach to test NN verifiers and examine their soundness, by developing a benchmark containing instances with *hidden counterexamples* – predefined counterexamples for the properties being verified, which can be kept away from NN verifier developers. NN verifiers often contain a preliminary check for counterexamples using adversarial attacks (Madry et al., 2018) before starting the more expensive verification process. Therefore, we aim to eliminate any easily found counterexamples and make the pre-defined counterexamples remain hidden against adversarial attacks, so that the NN verifiers enter the actual verification process. We then check whether the NN verifier falsely claims successful verification when we know a hidden counterexample exists.

We focus on a classification task using NNs, where the properties being verified concern whether the NN always makes robust predictions given small input perturbations within a pre-defined range. This is a commonly used setting for benchmarking NN verifiers. To build a benchmark with hidden counterexamples, we essentially need to train NNs that 1) make wrong predictions on the pre-defined counterexamples and 2) make correct and robust predictions on most other inputs so that counterexamples cannot be trivially found by adversarial attacks. To meet both objectives, we propose training NNs with a two-part loss function. The first part aims to achieve misclassification on the pre-defined counterexamples, while the second part is based on adversarial training (Madry et al., 2018) to enforce robustness on other inputs. Since there is an inherent conflict between these two objectives, making training particularly challenging, we strengthen the training by designing a margin objective in the first part and incorporating a perturbation sliding window in the second part.

Through this approach, our proposed training framework enables us to build a benchmark, named **SoundnessBench**, which incorporates 26 distinct NN models with diverse architectures (CNN, MLP and ViT), activation functions (ReLU, Sigmoid and Tanh), and perturbation radii, designed to comprehensively test the soundness of different verifiers. In these models, we planted a total of 206 hidden counterexamples. Using our SoundnessBench, we successfully identify real bugs in well-established and state-of-the-art NN verifiers including $\alpha,\beta$-CROWN, Marabou, and NeuralSAT.

## 2 Background and Related Work

**The problem of NN verification.** A NN can generally be viewed as a function $\boldsymbol{f} : \mathbb{R}^d \to \mathbb{R}^K$ that takes a $d$-dimensional input $\boldsymbol{x} \in \mathbb{R}^d$ and outputs $\boldsymbol{f}(\boldsymbol{x}) \in \mathbb{R}^K$. For instance, if the NN is for a $K$-way classification task, $\boldsymbol{f}(\boldsymbol{x})$ denotes the predicted logit score for each of the $K$ classes. NN verification involves a specification on the input and the output. The input can be specified by a region $\mathcal{C}$, and the properties to verify for the output can be specified using a function $s : \mathbb{R}^K \to \mathbb{R}$ that maps the output of the NN to a value, where the desired property is $h(\boldsymbol{x}) = s(\boldsymbol{f}(\boldsymbol{x})) > 0$. Formally, the NN verification problem can be defined as verifying:

$$\forall \boldsymbol{x} \in \mathcal{C}, \ h(\boldsymbol{x}) > 0, \quad \text{where } h(\boldsymbol{x}) = s(\boldsymbol{f}(\boldsymbol{x})). \tag{1}$$

**Soundness of NN verifiers.** An NN verifier is *sound* if every time the verifier claims that Eq. (1) holds, there exists no counterexample $\boldsymbol{x} \in \mathcal{C}$ that violates the condition $h(\boldsymbol{x}) > 0$. Even if an NN verifier is theoretically designed to be sound and rigorous, its implementation may contain bugs or errors that can compromise the soundness in practice. An NN verifier is *unsound* if the verifier claims that Eq. (1) holds but a counterexample $\boldsymbol{x} \in \mathcal{C}$ is found to have $h(\boldsymbol{x}) \leq 0$ and thus violate the desired property $h(\boldsymbol{x}) > 0$.

Although adversarial attacks such as projected gradient descent (PGD) (Madry et al., 2018) can often be used to find adversarial examples as counterexamples, in many cases when counterexamples cannot be found, checking the soundness of an NN verifier is challenging. In this work, we address the challenge by building a benchmark with pre-defined hidden counterexamples to test the soundness of NN verifiers.

For improving the soundness of NN verifiers, tools like DelBugV (Elsaleh & Katz, 2023) have been proposed for debugging NN verifiers, by shrinking bug-triggering networks and isolating the minimal piecewise-linear structure responsible for soundness violations. However, it can only be used once concrete test cases triggering bugs have been found. Proof-producing verifiers (Isac et al., 2022) produce independently checkable proofs for each claimed verification verdict. However, they are restricted to a specific base verifier and cannot test the soundness of most other independently developed NN verifiers. In contrast, our work is designed for testing the soundness of arbitrary NN verifiers and triggering new bugs with our test cases. Furthermore, some other works focused on subtle soundness issues in NN verifiers, namely issues from floating point numerical errors (Zombori et al., 2021; Jia & Rinard, 2021; Szász et al., 2025), which are orthogonal to our focus on soundness issues beyond numerical errors.

**NN verification benchmarks.** Previous works have proposed numerous benchmarks for evaluating NN verifiers. There have been efforts on standardizing the benchmarking for NN verification (Tacchella et al., 2019; Shriver et al., 2021). Subsequently, multiple iterations of VNN-COMP have compiled comprehensive benchmarks in the competitions with detailed descriptions of the benchmarks in the competition reports (Bak et al., 2021; Müller et al., 2022; Brix et al., 2023a; 2024). The existing benchmarks cover diverse NN architectures, such as fully-connected NNs, convolutional neural networks (CNNs), Transformers (Vaswani et al., 2017; Dosovitskiy et al., 2021; Shi et al., 2020; 2025), as well as many complicated NN-based models used in applications such as dataset indexing and cardinality prediction (He et al., 2022), power systems (Guha et al., 2019), and controllers (Yang et al., 2024). The existing benchmarks and VNN-COMP commonly rely on cross-validating the results by different verifiers to check the soundness of the verifiers. When only some verifier claims a successful verification on a property while no other verifier can falsify the property, the soundness of the verifier cannot be readily tested. In this work, we aim to build a novel benchmark containing pre-defined counterexamples to test the soundness of NN verifiers, without requiring cross-validation among different verifiers.

**Adversarial training.** Our work leverages adversarial training (Goodfellow et al., 2015; Madry et al., 2018), to keep our pre-defined counterexamples hidden to adversarial attacks and eliminate trivial counterexamples that can be easily found by adversarial attacks. Adversarial training was originally proposed for defending against adversarial attacks and it essentially solves a min-max problem:

$$\min_{\boldsymbol{\theta}} \mathbb{E}_{(\boldsymbol{x},y) \sim \mathcal{D}} \left[ \max_{\|\boldsymbol{\delta}\|_\infty \leq \epsilon} \mathcal{L}_{CE}(\boldsymbol{f_\theta}(\boldsymbol{x} + \boldsymbol{\delta}), y) \right], \tag{2}$$

where the model parameters $\boldsymbol{\theta}$ are optimized to minimize the cross entropy loss function $\mathcal{L}_{\text{CE}}$, under an approximately worst-case perturbation $\boldsymbol{\delta}$ ($\|\boldsymbol{\delta}\|_\infty \leq \epsilon$) found by an adversarial attack such as FGSM (Goodfellow et al., 2015) or PGD (Madry et al., 2018). In our method, we incorporate an adversarial training objective to make the NN robust for most inputs except for the pre-defined hidden counterexamples.

**Planting counterexamples.** Planting counterexamples (or adversarial examples) has also been studied in Zimmermann et al. (2022) but with a different purpose and method. They aimed to test adversarial attacks and identify weak attacks and thus weak defense evaluations. Hence they did not consider making planted counterexamples hidden against adversarial attacks. In contrast, we aim to not only plant pre-defined counterexamples, but also make such counterexamples hidden against adversarial attacks, which requires a new design for the training to achieve our more challenging objective, and we further discuss methodological differences in Section 3. Additionally, another line of works studied backdoor attacks on NNs (Gu et al., 2017; Liu et al., 2018), which injected wrong behaviors into NNs when a trigger presents in the input, sharing some similarity with our hidden counterexamples. Unlike these works, since our pre-defined counterexamples are for NN verification tested on individual instances, they are generated for each instance, as opposed to a trigger for the entire model in backdoor attacks. Meanwhile, we require the model to be mostly robust on the input region that contains the pre-defined counterexample for each instance, to keep the counterexample hidden against adversarial attacks, which is also new compared to backdoor attacks.

## 3 Methodology

### 3.1 Problem Definition

Our problem concerns building a benchmark for NN verification with hidden pre-defined counterexamples. Our benchmark contains a set of *instances*, where each instance $(\boldsymbol{f}, \boldsymbol{x}_0, y, \epsilon)$ denotes a property of the NN model $\boldsymbol{f}$ to be verified.

**NN verification for $K$-way classification.** In our work, we focus on a verification problem in a $K$-way classification that is commonly used to benchmark NN verifiers. The verification problem Eq. (1) is specified as testing the robustness of the model against small perturbations to the input. The input region is defined as a small $\ell_\infty$ ball with a perturbation radius of $\epsilon$ around a clean data point $\boldsymbol{x}_0$: $\mathcal{C} = \mathcal{B}(\boldsymbol{x}_0, \epsilon) := \{\boldsymbol{x} \mid \|\boldsymbol{x} - \boldsymbol{x}_0\|_\infty \leq \epsilon\}$, where $\boldsymbol{x} - \boldsymbol{x}_0$ is a small perturbation within the allowed range. The output specification can be defined as $s(\boldsymbol{z}) := \min\{\boldsymbol{z}_y - \boldsymbol{z}_i \mid 1 \leq i \leq K, i \neq c\}$ for $\boldsymbol{z} = \boldsymbol{f}(\boldsymbol{x})$, where $y$ is the ground-truth label for the classification, $i \neq y$ is any other incorrect class, and $s(\boldsymbol{z})$ denotes the minimum margin between the ground-truth class and other incorrect classes. The goal of verification is to verify that $\boldsymbol{f}_y(\boldsymbol{x}) - \boldsymbol{f}_i(\boldsymbol{x}) > 0$, for any input $\boldsymbol{x}$ within the $\ell_\infty$ ball $\mathcal{B}(\boldsymbol{x}_0, \epsilon)$ and any other class $i$, meaning that the classification is always correct and thus robust under any small perturbation with $\|\boldsymbol{x} - \boldsymbol{x}_0\|_\infty \leq \epsilon$, as:

$$\forall \boldsymbol{x} \in \mathcal{B}(\boldsymbol{x}_0, \epsilon), \; h(\boldsymbol{x}) = \min\{\boldsymbol{f}_y(\boldsymbol{x}) - \boldsymbol{f}_i(\boldsymbol{x}) \mid 1 \leq i \leq K, i \neq y\} > 0, \tag{3}$$

**Unverifiable instances with pre-defined counterexamples.** Since we are primarily concerned with testing the soundness of NN verifiers, our benchmark contains *unverifiable instances* with pre-defined counterexamples deliberately planted. Specifically, for an instance $(\boldsymbol{f}, \boldsymbol{x}_0, y, \epsilon)$, we aim to plant a counterexample $\boldsymbol{x}_{\text{CEX}} = \boldsymbol{x}_0 + \boldsymbol{\delta}_{\text{CEX}}$ with perturbation $\boldsymbol{\delta}_{\text{CEX}}$ and an incorrect label $y_{\text{CEX}}$ ($y_{\text{CEX}} \neq y$), such that Eq. (3) is violated:

$$\boldsymbol{x}_{\text{CEX}} = \boldsymbol{x}_0 + \boldsymbol{\delta}_{\text{CEX}} \quad \text{s.t.} \; \|\boldsymbol{\delta}_{\text{CEX}}\|_\infty \leq \epsilon, \; \boldsymbol{f}_y(\boldsymbol{x}_{\text{CEX}}) - \boldsymbol{f}_{y_{\text{CEX}}}(\boldsymbol{x}_{\text{CEX}}) \leq 0, \tag{4}$$

where "CEX" is short for "counterexample". Such an instance allows us to detect bugs in an NN verifier if it claims Eq. (3) can be verified while we know that $\boldsymbol{x}_{\text{CEX}}$ satisfying Eq. (4) exists.

**Making pre-defined counterexamples hidden.** Additionally, we aim to make the planted counterexample "hidden" to NN verifiers so that NN verifiers cannot easily falsify the unverifiable instances without even entering the actual verification procedure. In particular, we aim to make the counterexample hard to find by adversarial attacks which are often used in NN verifiers as a preliminary check before using a more

expensive verification procedure. Suppose $\mathcal{A}(\boldsymbol{x}_0, \epsilon, y, i)$ is an adversarial attack such as PGD (Madry et al., 2018) solving the following problem:

$$\underset{\|\boldsymbol{\delta}\|_\infty \leq \epsilon}{\arg\min} \; \boldsymbol{f}_y(\boldsymbol{x}_0 + \boldsymbol{\delta}) - \boldsymbol{f}_i(\boldsymbol{x}_0 + \boldsymbol{\delta}), \tag{5}$$

where the solution returned by the adversarial attack usually does not reach the global minimum. We aim to achieve:

$$\forall i \neq y, \;\; \boldsymbol{f}_y(\boldsymbol{x}_0 + \mathcal{A}(\boldsymbol{x}_0, \epsilon, y, i)) - \boldsymbol{f}_i(\boldsymbol{x}_0 + \mathcal{A}(\boldsymbol{x}_0, \epsilon, y, i)) > 0, \tag{6}$$

which means the example $\boldsymbol{x}_0 + \mathcal{A}(\boldsymbol{x}_0, \epsilon, y, i)$ found by the adversarial attack is not a counterexample.

If both Eq. (4) and Eq. (6) are achieved, we call $\boldsymbol{x}_{\mathrm{CEX}}$ a **hidden counterexample**, and we have ground-truth for the instance $(\boldsymbol{f}, \boldsymbol{x}_0, y, \epsilon)$, knowing that the instance cannot be verified.

## 3.2 Overall Pipeline

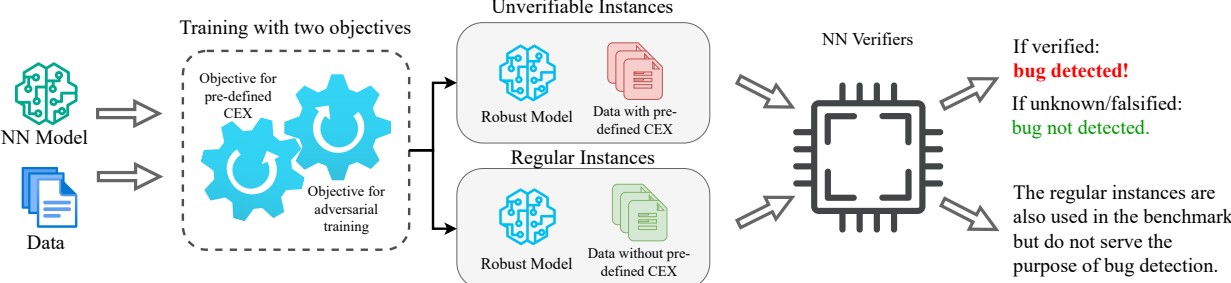

Figure 1: A high-level overview of our SoundnessBench and its purpose. NN models are trained on a synthetic dataset using the two-objective training framework described in Section 3.4, to produce both unverifiable and regular instances. Unverifiable instances are then used to test the soundness of various NN verifiers, while regular instances are also added to prevent developers of NN verifiers from directly knowing all the instances are unverifiable. A bug is detected if an NN verifier claims a successful verification for any unverifiable instance.

Our benchmark is constructed with a number of different settings. Each setting is identified by: 1) A dataset that provides original examples $\boldsymbol{x}_0$ and ground-truth labels $y$; 2) A neural network architecture for model $\boldsymbol{f}$; 3) The perturbation size $\epsilon$. For each setting, our pipeline proceeds as follows: For each example from the original dataset, we generate a random target class. We aim to make half of these examples unverifiable with hidden counterexamples (i.e., unverifiable instances), by generating a random pre-defined counterexample. The other half are regular examples without pre-defined counterexamples, so that these examples are potentially verifiable and NN verifier developers do not simply know all the examples are unverifiable. We then train a model with both an adversarial training objective and an additional objective for producing counterexamples. After the training, for the examples with pre-defined counterexamples, we filter out those with counterexamples that can be found by a stronger adversarial attack, or where the pre-defined counterexamples do not end up becoming counterexamples. We discuss these stages in more detail below. We now focus on a given setting with model $\boldsymbol{f}$ and perturbation size $\epsilon$, and we omit $\boldsymbol{f}$ and $\epsilon$ when describing instances below. Figure 1 provides an overview.

## 3.3 Data generation

For each setting, we construct a dataset $\mathbb{D} = \mathbb{D}_{\mathrm{unv}} \cup \mathbb{D}_{\mathrm{regular}}$ consisting of $n$ unverifiable instances and $n$ regular instances, respectively:

$$\mathbb{D} = \mathbb{D}_{\mathrm{unv}} \cup \mathbb{D}_{\mathrm{regular}} = \{(\boldsymbol{x}_0^{(1)}, y^{(1)}), (\boldsymbol{x}_0^{(2)}, y^{(2)}), \cdots, (\boldsymbol{x}_0^{(2n)}, y^{(2n)})\}, \tag{7}$$

$$\mathbb{D}_{\mathrm{unv}} = \{(\boldsymbol{x}_0^{(\mathrm{unv},1)}, y^{(\mathrm{unv},1)}), (\boldsymbol{x}_0^{(\mathrm{unv},2)}, y^{(\mathrm{unv},2)}), \cdots, (\boldsymbol{x}_0^{(\mathrm{unv},n)}, y^{(\mathrm{unv},n)})\}, \tag{8}$$

$$\mathbb{D}_{\text{regular}} = \{(\boldsymbol{x}_0^{(\text{regular},1)}, y^{(\text{regular},1)}), (\boldsymbol{x}_0^{(\text{regular},2)}, y^{(\text{regular},2)}), \cdots, (\boldsymbol{x}_0^{(\text{regular},n)}, y^{(\text{regular},n)})\}. \qquad (9)$$

In these instances, $\boldsymbol{x}_0$ and $y$ come from an original dataset, where the first $n$ examples are used to construct $\mathbb{D}_{\text{unv}}$ while the remaining $n$ examples are used for $\mathbb{D}_{\text{regular}}$. For each unverifiable instance $(\boldsymbol{x}_0^{(\text{unv},i)}, y^{(\text{unv},i)}) \in \mathbb{D}_{\text{unv}}$ $(1 \leq i \leq n)$, we generate a pre-defined counterexample $(\boldsymbol{x}_{\text{CEX}}^{(i)}, y_{\text{CEX}}^{(i)})$, where $\boldsymbol{x}_{\text{CEX}}^{(i)} = \boldsymbol{x}_0^{(\text{unv},i)} + \boldsymbol{\delta}_{\text{CEX}}^{(i)}$, such that $\boldsymbol{\delta}_{\text{CEX}}^{(i)}$ is the perturbation and $y_{\text{CEX}}^{(i)}$ is the target label. $\boldsymbol{\delta}_{\text{CEX}}^{(i)}$ is randomly generated from $([-\epsilon, -r \cdot \epsilon] \bigcup [r \cdot \epsilon, \epsilon])^d$ $(0 \leq r < 1)$ satisfying $r \cdot \epsilon \leq \|\boldsymbol{\delta}_{\text{CEX}}^{(i)}\|_\infty \leq \epsilon$, and $y_{\text{CEX}}^{(i)}$ is randomly generated from $\{j \mid 1 \leq j \leq K, j \neq y^{(\text{unv},i)}\}$ satisfying $y_{\text{CEX}}^{(i)} \neq y^{(\text{unv},i)}$. Here, since we aim to ensure $\boldsymbol{x}_0^{(\text{unv},i)}$ and $\boldsymbol{x}_{\text{CEX}}^{(i)}$ have different labels, we make them separated by setting $\|\boldsymbol{\delta}_{\text{CEX}}^{(i)}\|_\infty \geq r \cdot \epsilon$ with a hyperparameter $r$ $(0 \leq r < 1)$ for the minimum distance, where $r$ is set to a value close to 1. This is similar to how Zimmermann et al. (2022) controlled the distance and generated perturbed examples near the boundary of the $\mathcal{B}(\boldsymbol{x}_0, \epsilon)$ perturbation ball when creating their dataset for planting counterexamples.

### 3.4   Training

Our training comprises two objectives. The first part aims to make our pre-defined counterexamples become real counterexamples, to satisfy Eq. (4). The second part aims to eliminate counterexamples that can be found by adversarial attacks to satisfy Eq. (6) for both unverifiable instances and regular instances. We explain these two objectives below:

**Objective for the pre-defined counterexamples.** In the first objective regarding Eq. (4), for each unverifiable instance $(\boldsymbol{x}_0^{(\text{unv},i)}, y^{(\text{unv},i)})$ $(1 \leq i \leq n)$ with the pre-defined counterexample $(\boldsymbol{x}_{\text{CEX}}^{(i)}, y_{\text{CEX}}^{(i)})$, we aim to make the class predicted from output logits $\boldsymbol{f}(\boldsymbol{x}_{\text{CEX}}^{(i)})$ match target label $y_{\text{CEX}}^{(i)}$, i.e., $\arg\max_{1 \leq j \leq K} \boldsymbol{f}_j(\boldsymbol{x}_{\text{CEX}}^{(i)}) = y_{\text{CEX}}^{(i)}$. This can be achieved by a standard cross-entropy (CE) loss. However, it tends to maximize $\boldsymbol{f}_{y_{\text{CEX}}^{(i)}}(\boldsymbol{x}_{\text{CEX}}^{(i)})$ and minimize $\boldsymbol{f}_{y^{(\text{unv},i)}}(\boldsymbol{x}_{\text{CEX}}^{(i)})$ without a cap on the margin between them. We find that making $\boldsymbol{f}_{y_{\text{CEX}}^{(i)}}(\boldsymbol{x}_{\text{CEX}}^{(i)})$ substantially greater than $\boldsymbol{f}_{y^{(\text{unv},i)}}(\boldsymbol{x}_{\text{CEX}}^{(i)})$ tends to make the counterexamples more obvious to adversarial attacks and thus makes our training with the second objective difficult. Therefore, we design a *margin objective* instead to achieve $0 \leq \boldsymbol{f}_{y_{\text{CEX}}^{(i)}}(\boldsymbol{x}_{\text{CEX}}^{(i)}) - \boldsymbol{f}_{y^{(\text{unv},i)}}(\boldsymbol{x}_{\text{CEX}}^{(i)}) \leq \lambda$ with a maximum desired margin $\lambda$:

$$\mathcal{L}_{\text{CEX}} = \frac{1}{n} \sum_{i=1}^{n} \max\{0, \boldsymbol{f}_{y^{(\text{unv},i)}}(\boldsymbol{x}_{\text{CEX}}^{(i)}) - \boldsymbol{f}_{y_{\text{CEX}}^{(i)}}(\boldsymbol{x}_{\text{CEX}}^{(i)}) + \lambda)\}. \qquad (10)$$

This is different from Zimmermann et al. (2022) as they only trained a linear classification head to separate the frozen features of perturbed examples inside the perturbation ball and on the boundary of the perturbation ball, thereby making the examples on the boundary become counterexamples. In contrast, we train the whole model, as we additionally have a more challenging objective of making the counterexamples hidden against adversarial attacks (achieved via the adversarial training objective described below), and we achieve the objective for the pre-defined counterexamples here with a margin objective.

**Objective with adversarial training.** Our second objective is adversarial training as defined in Eq. (2), applied to the entire dataset $\mathbb{D}$ with both unverifiable instances and regular instances. However, since by default a new perturbation $\boldsymbol{\delta}^{(i)}$ $(1 \leq i \leq 2n)$ is generated at each training epoch for every example, we find that the training struggles to converge, if we only use the new perturbation. This is because training on a new perturbation can often break the model on other perturbations including those used in previous epochs, when we have a conflicting objective for pre-defined counterexamples. To mitigate this issue, we propose a *perturbation sliding window*, where we use multiple perturbations generated from the most recent epochs. For each instance $(\boldsymbol{x}_0^{(i)}, y^{(i)})$, we maintain a window $\mathbb{W}^{(i)}$ of perturbations generated in the most recent $w$ epochs: $\mathbb{W}^{(i)} = \{\boldsymbol{\delta}_1^{(i)}, \boldsymbol{\delta}_2^{(i)}, \cdots, \boldsymbol{\delta}_w^{(i)}\}$, where $w$ is the maximum length of the sliding window (there can be fewer than $w$ elements in $\mathbb{W}^{(i)}$ in the first $w-1$ epochs). Using this window, our objective that utilizes all

the perturbations in the sliding window is:

$$\mathcal{L}_{\text{adv}} = \mathcal{L}_{\text{adv, window}} = \frac{1}{2n} \sum_{1 \leq i \leq 2n} \frac{1}{|\mathbb{W}^{(i)}|} \sum_{\boldsymbol{\delta} \in \mathbb{W}^{(i)}} \mathcal{L}_{\text{CE}}(\boldsymbol{f}(\boldsymbol{x}_0^{(i)} + \boldsymbol{\delta}), y^{(i)}). \tag{11}$$

This differs significantly from Zimmermann et al. (2022) which did not have adversarial training to make counterexamples hidden.

**Overall objective.** Suppose model $\boldsymbol{f}$ is parameterized by parameters $\boldsymbol{\theta}$, denoted as $\boldsymbol{f_\theta}$, and our two loss functions both depend on $\boldsymbol{\theta}$, denoted as $\mathcal{L}_{\text{CEX}}(\boldsymbol{\theta})$ and $\mathcal{L}_{\text{adv}}(\boldsymbol{\theta})$. Then our overall loss function combines the two objectives as:

$$\mathcal{L}(\boldsymbol{\theta}) = \mathcal{L}_{\text{CEX}}(\boldsymbol{\theta}) + \mathcal{L}_{\text{adv}}(\boldsymbol{\theta}), \tag{12}$$

and we optimize model parameters by solving $\arg\min_{\boldsymbol{\theta}} \mathcal{L}(\boldsymbol{\theta})$ using a gradient-based optimizer. The complete training algorithm is shown in Algorithm 1 in Appendix A.

### 3.5 Benchmark Design

Table 1: List of model architectures included in our benchmark. "Conv $k \times w \times h$" represents a 2D convolutional layer with $k$ filters of kernel size $w \times h$, "FC $w$" represents a fully-connected layer with $w$ hidden neurons, "AvgPool $k \times k$" represents a 2D average pooling layer with kernel size $k \times k$. An activation function is added after every convolutional layer or fully-connected layer.

| Name | Model Architecture | Activation Function |
|---|---|---|
| CNN 1 Conv | Conv $10 \times 3 \times 3$, FC 1000, FC 100, FC 20, FC 2 | ReLU |
| CNN 2 Conv | Conv $5 \times 3 \times 3$, Conv $10 \times 3 \times 3$, FC 1000, FC 100, FC 20, FC 2 | ReLU |
| CNN 3 Conv | Conv $5 \times 3 \times 3$, Conv $10 \times 3 \times 3$, Conv $20\ 3 \times 3$, FC 1000, FC 100, FC 20, FC 2 | ReLU |
| CNN AvgPool | Conv $10 \times 3 \times 3$, AvgPool $3 \times 3$, FC 1000, FC 100, FC 20, FC 2 | ReLU |
| MLP 4 Hidden | FC 100, FC 1000, FC 1000, FC 1000, FC 20, FC 2 | ReLU |
| MLP 5 Hidden | FC 100, FC 1000, FC 1000, FC 1000, FC 1000, FC 20, FC 2 | ReLU |
| CNN Tanh | Conv $10 \times 3 \times 3$, FC 1000, FC 100, FC 20, FC 2 | Tanh |
| CNN Sigmoid | Conv $10 \times 3 \times 3$, FC 1000, FC 100, FC 20, FC 2 | Sigmoid |
| ViT | Modified ViT (Shi et al., 2025) with patch size $1 \times 1$, 2 attention heads and embedding size 16 | ReLU |

**Scale.** We focus on using relatively small models and low-dimensional synthetic data. Since the goal of our benchmark is to test the soundness of NN verifiers rather than benchmark performance, our models and data do not have to be large-scale or as practical as real-world ones. We may draw an analogy to fuzzing techniques in software engineering, where simple and synthetic but specifically constructed inputs, rather than complicated real-world inputs, are used for testing software and identifying bugs. Additionally, large-scale models and data can significantly increase the difficulty for NN verifiers, and NN verifiers can fail to provide meaningful verification results or simply fail to support complicated cases, without exposing bugs. Verifiers typically require relaxation, and on relatively larger settings, looser relaxation of other correct layers can compensate for the wrong bound computation, making the final verification more conservative. Although NN verifiers can often verify much larger models in other benchmarks, they depend on models being specifically trained, and our models trained with hidden counterexamples are more challenging to verify. As shown in Table 6, models and data at our small scale are already difficult for current NN verifiers. Therefore, by using relatively small models and data, we allow more verifiers to be tested against our benchmark to maximize bug exposure. Nevertheless, we also show results on MNIST in Appendix E, where verification is much more difficult.

**Model.** As shown in Table 1, we build our benchmark to contain diverse NN architectures and activation functions. We incorporate 9 distinct NN architectures, spanning MLPs (NNs with fully-connected layers), CNNs, and ViTs, with different activation functions including ReLU, Tanh, and Sigmoid. The models are designed to have moderate size for existing NN verifiers to handle, and in total, there are 26 models in our benchmark.

**Data.** Our dataset of original examples $(\boldsymbol{x}_0^{(1)}, y^{(1)}), (\boldsymbol{x}_0^{(2)}, y^{(2)}), \cdots, (\boldsymbol{x}_0^{(n)}, y^{(n)})$ is synthetic and randomly generated following uniform distributions, where $\boldsymbol{x}_0^{(i)}$ is sampled from $[-1, 1]^d$ and $y^{(i)}$ is sampled from $\{0, 1\}$ for $1 \leq i \leq 2n$. We use various input sizes and perturbation radii, as shown in Table 2. Specifically, for MLP models, we use 1D data with input size $d = 10$; for other models, we use 2D data with input sizes $d = 1 \times 5 \times 5$ and $d = 3 \times 5 \times 5$, respectively, where 1 and 3 are the numbers of input channels considered by CNNs and ViTs. For the perturbation radii, we use $\epsilon = 0.2$ and $\epsilon = 0.5$. For each model with a particular perturbation radius and input size, we set $n = 10$ as the number of instances and generate 10 unverifiable instances $(\boldsymbol{x}_0^{(\text{unv},i)}, y^{(\text{unv},i)}) \in \mathbb{D}_{\text{unv}}$ $(1 \leq i \leq n)$, each with a corresponding pre-defined counterexample $(\boldsymbol{x}_{\text{CEX}}^{(i)}, y_{\text{CEX}}^{(i)})$, and 10 regular instances $(\boldsymbol{x}_0^{(\text{regular},i)}, y^{(\text{regular},i)}) \in \mathbb{D}_{\text{regular}}$ $(1 \leq i \leq n)$. We set $r = 0.98$ for $\|\boldsymbol{\delta}_{\text{CEX}}^{(i)}\|_\infty \geq r \cdot \epsilon$, introduced in Section 3.3, so that each pre-defined counterexample is close to the boundary of the input perturbation ball. During the data generation, to ensure that different examples are sufficiently separated, we require that there is no intersection between the perturbation balls of different examples and retry the generation until no intersection exists. After the training, for the unverifiable instances, we only keep those with hidden counterexamples that are not found by strong adversarial attacks conducted in Section 4.2. Thus, there can be fewer than 10 unverifiable instances if any pre-defined counterexample is found by an adversarial attack.

We compare settings with existing benchmarks in Appendix B.

## 4 Experiments

In this section, we conduct comprehensive experiments. We first build our SoundnessBench by our proposed method with implementation details provided in Section 4.1, we evaluate the effectiveness of our training in Section 4.2. We then use our benchmark to test the soundness of several state-of-the-art NN verifiers in Section 4.3. Additionally, in Section 4.4, we conduct an ablation study to justify the design choices for our training. We also provide additional results in the appendix about testing NN verifiers with synthetic bugs (Appendix C), repeating our experiments with multiple random seeds (Appendix D), and experimenting on MNIST (Appendix E).

### 4.1 Implementation Details

For every model in the benchmark, we apply our aforementioned training framework with the following hyperparameters: (1) Number of training epochs is set to 5000; (2) Adam optimizer (Kingma & Ba, 2015) with a cyclic learning rate schedule is used, where the learning rate gradually increases from 0 to a peak value during the first half of the training, and then it gradually decreases to 0 during the second half of the training, with the peak learning rate set to 0.001; (3) The threshold in the margin objective defined in Eq. (10) is set to $\lambda = 0.01$; (4) The size of the perturbation sliding window used in Eq. (11) is set to $w = 300$; (5) We use the PGD attack (Madry et al., 2018) for the adversarial training with up to 150 restarts (number of different random initialization for the PGD attack) and 150 attack steps.

### 4.2 Training NNs and Building the Benchmark

**Evaluation method.** After the models are trained, we first evaluate we have successfully planted hidden counterexamples. For each unverifiable instance $(\boldsymbol{x}_0^{(\text{unv},i)}, y^{(\text{unv},i)})$ $(1 \leq i \leq n)$, we evaluate it in three steps:

1. We check whether the model predicts the correct class $y^{(\text{unv},i)}$ for unperturbed input $\boldsymbol{x}_0^{(\text{unv},i)}$;
2. We check whether the model predicts the expected class $y_{\text{CEX}}^{(i)} \neq y^{(\text{unv},i)}$ for the pre-defined counterexample $\boldsymbol{x}_{\text{CEX}}^{(i)}$;
3. We check whether no counterexample can be found by adversarial attacks in the region $\mathcal{B}(\boldsymbol{x}_0^{(\text{unv},i)}, \epsilon)$, i.e. $\boldsymbol{x}_{\text{CEX}}^{(i)}$ is a hidden counterexample.

For the regular instances $(\boldsymbol{x}_0^{(\text{regular},i)}, y^{(\text{regular},i)})$ $(1 \leq i \leq n)$, we evaluate them in two steps:

Table 2: Evaluation results of trained models. Our goal is to achieve high "hidden CEX" (up to 10) for unverifiable instances and high "No CEX found" for regular instances.

| Training Settings | | | Results for Unverifiable Instances | | | Results for Regular Instances | |
|---|---|---|---|---|---|---|---|
| Name | $\epsilon$ | Input size | Correct $\boldsymbol{x}_0^{(\mathrm{unv},i)}$ | Incorrect $\boldsymbol{x}_{\mathrm{CEX}}^{(i)}$ | Hidden CEX | Correct $\boldsymbol{x}_0^{(\mathrm{regular},i)}$ | No CEX found |
| CNN 1 Conv | 0.2 | $1 \times 5 \times 5$ | 10 | 10 | **9** | 10 | 10 |
| CNN 1 Conv | 0.2 | $3 \times 5 \times 5$ | 10 | 10 | **9** | 10 | 10 |
| CNN 1 Conv | 0.5 | $1 \times 5 \times 5$ | 10 | 10 | **10** | 10 | 10 |
| CNN 1 Conv | 0.5 | $3 \times 5 \times 5$ | 10 | 10 | **8** | 10 | 10 |
| CNN 2 Conv | 0.2 | $1 \times 5 \times 5$ | 10 | 10 | **7** | 10 | 10 |
| CNN 2 Conv | 0.2 | $3 \times 5 \times 5$ | 10 | 10 | **7** | 10 | 10 |
| CNN 2 Conv | 0.5 | $1 \times 5 \times 5$ | 10 | 10 | **10** | 10 | 10 |
| CNN 2 Conv | 0.5 | $3 \times 5 \times 5$ | 10 | 10 | **10** | 10 | 10 |
| CNN 3 Conv | 0.2 | $1 \times 5 \times 5$ | 10 | 10 | **10** | 10 | 10 |
| CNN 3 Conv | 0.2 | $3 \times 5 \times 5$ | 10 | 10 | **7** | 10 | 10 |
| CNN 3 Conv | 0.5 | $1 \times 5 \times 5$ | 10 | 10 | **9** | 10 | 10 |
| CNN 3 Conv | 0.5 | $3 \times 5 \times 5$ | 10 | 10 | **10** | 10 | 10 |
| CNN AvgPool | 0.2 | $1 \times 5 \times 5$ | 8 | 10 | **0** | 10 | 10 |
| CNN AvgPool | 0.2 | $3 \times 5 \times 5$ | 10 | 10 | **1** | 10 | 10 |
| CNN AvgPool | 0.5 | $1 \times 5 \times 5$ | 10 | 10 | **9** | 10 | 10 |
| CNN AvgPool | 0.5 | $3 \times 5 \times 5$ | 10 | 10 | **10** | 10 | 10 |
| MLP 4 Hidden | 0.2 | 10 | 10 | 10 | **9** | 10 | 10 |
| MLP 4 Hidden | 0.5 | 10 | 10 | 10 | **9** | 10 | 10 |
| MLP 5 Hidden | 0.2 | 10 | 10 | 10 | **4** | 10 | 10 |
| MLP 5 Hidden | 0.5 | 10 | 10 | 10 | **9** | 10 | 10 |
| CNN Tanh | 0.2 | $1 \times 5 \times 5$ | 10 | 10 | **8** | 10 | 10 |
| CNN Tanh | 0.2 | $3 \times 5 \times 5$ | 10 | 10 | **10** | 10 | 10 |
| CNN Sigmoid | 0.2 | $1 \times 5 \times 5$ | 10 | 10 | **1** | 10 | 10 |
| CNN Sigmoid | 0.2 | $3 \times 5 \times 5$ | 10 | 10 | **10** | 10 | 10 |
| ViT | 0.2 | $1 \times 5 \times 5$ | 10 | 10 | **10** | 10 | 10 |
| ViT | 0.2 | $3 \times 5 \times 5$ | 10 | 10 | **10** | 10 | 10 |

1. We check whether the model predicts the correct class $y^{(\mathrm{regular},i)}$ for unperturbed input $\boldsymbol{x}_0^{(\mathrm{regular},i)}$;
2. We check whether no counterexample can be found by adversarial attacks in the region $\mathcal{B}(\boldsymbol{x}_0^{(\mathrm{regular},i)}, \epsilon)$.

To make sure that our pre-defined counterexamples are truly hidden against adversarial attacks that try to find counterexamples, we evaluate them using strong adversarial attacks. Specifically, we apply PGD attack with strong attack parameters, 1000 restarts and 5000 PGD steps, and also AutoAttack (Croce & Hein, 2020) which combines multiple adversarial attack strategies. For the pre-defined counterexamples, only those passing all of these attacks are considered to be hidden counterexamples.

**Results.** Table 2 presents the evaluation results. For most unverifiable instances, the model makes a correct prediction on the unperturbed input while making an incorrect prediction on the pre-defined counterexample as expected. When we require the pre-defined counterexamples to be hidden against adversarial attacks, a large number of hidden counterexamples are produced, and notably, 22 of the 26 models have at least 7 hidden counterexamples. These results demonstrate the effectiveness of our method in producing hidden counterexamples that we use to test the soundness of verifiers. Meanwhile, for regular instances, all the models achieve correct predictions on the unperturbed input and no counterexample is found. This shows that the models have robust performance on the instances where pre-defined counterexamples are not added.

**Location of counterexamples.** To better understand why our counterexamples are difficult to find, we analyze the location of the counterexamples with respect to the decision boundary. For each pre-defined counterexample, using PGD attack, we approximately find the closest point in the input space that would

make the model produce the correct prediction, which measures how close the counterexample is to the decision boundary. We conduct the analysis on two models, CNN AvgPool ($\epsilon = 0.5$ and input size $1 \times 5 \times 5$) and ViT ($\epsilon = 0.2$ and input size $3 \times 5 \times 5$). The mean $\ell_\infty$ distances to the decision boundary are $2.6 \times 10^{-4}$ for the CNN AvgPool model and $7.3 \times 10^{-4}$ for the ViT model. The counterexamples tend to be very close to the decision boundary, thus making them hard to find by adversarial attacks.

### 4.3 Testing NN Verifiers

Table 3: Abbreviation of different versions of verifiers.

| Verifier | Abbreviation | Description |
|---|---|---|
| $\alpha,\beta$-CROWN | ABC-A | $\alpha,\beta$-CROWN using activation splitting |
| $\alpha,\beta$-CROWN | ABC-I | $\alpha,\beta$-CROWN using input splitting |
| NeuralSAT | NSAT-A | NeuralSAT using activation splitting |
| NeuralSAT | NSAT-I | NeuralSAT using input splitting |
| PyRAT | PyRAT | Default PyRAT |
| Marabou 2023 | MB-23 | Marabou used in VNN-COMP 2023 (branch `vnn-comp-23`) |
| Marabou 2024 | MB-24 | Marabou used in VNN-COMP 2024 (branch `vnn-comp-24`) |

We test the soundness of four representative NN verifiers, including $\alpha,\beta$-**CROWN**[1] (Zhang et al., 2018; Xu et al., 2020; 2021; Wang et al., 2021; Zhang et al., 2022; Shi et al., 2025), **NeuralSAT**[2] (Duong et al., 2024), **PyRAT**[3] (Lemesle et al., 2024), and **Marabou** (Katz et al., 2019; Wu et al., 2024). For $\alpha,\beta$-CROWN and NeuralSAT, we consider two distinct splitting strategies in their branch-and-bound for the verification, activation splitting and input splitting, which conduct branch-and-bound on the activation space and input space, respectively. For Marabou, we include two versions: Marabou 2023[4] and Marabou 2024[5], which were used in VNN-COMP 2023 and VNN-COMP 2024, respectively. We include Marabou 2023 because it supported ViT models, while Marabou 2024 does not. To streamline our experiment tables, we define abbreviations for different versions of verifiers, as listed in Table 3. A timeout of 100 seconds is applied on all the verification experiments, similar to what many existing benchmarks use (Bak et al., 2021; Müller et al., 2022; Brix et al., 2023a; 2024). If a verifier fails to output a result for an instance within the time limit, the instance is claimed as neither verified nor falsified.

**Main findings.** The results are presented in Tables 4 to 6, with the following findings:

1. **Our benchmark successfully identified bugs in three well-established verifiers: including $\alpha,\beta$-CROWN, NeuralSAT, and Marabou 2023.** Table 4 shows the proportion of "verified" instances claimed by the verifiers on *unverifiable* instances, measuring the ***unsoundness*** of verifiers. For the entries with *nonzero* values, the results show that these verifiers' claims are *unsound* and the verifiers contain bugs in these cases.

    Note that it is reasonable that no bug is found on most other models, because these well-developed verifiers are sound in most cases. Nevertheless, our benchmark enables researchers or developers of NN verifiers to sanity check their work in future development. When they make significant changes to existing verifiers or develop new verifiers that can support new settings, which is often error-prone, they can use our SoundnessBench. Alternatively, they can use our method to train new models with pre-defined counterexamples. SoundnessBench will help improve the quality and reliability of verifiers built in the community, and broadly impact safety-critical domains utilizing NN verifiers.

2. **Hidden counterexamples in our benchmark are difficult for existing verifiers to find.** Table 5 shows the proportion of unverifiable instances falsified by NN verifiers. The results show that most unverifiable instances are not falsified by the verifiers, and thus most of our hidden counterexamples successfully evade the falsification mechanisms in NN verifiers and remain hidden.

---

[1]`https://github.com/Verified-Intelligence/alpha-beta-CROWN_vnncomp2024`

[2]`https://github.com/dynaroars/neuralsat/commit/8c6f006c1acb293ce3f5a995935fd7bc1a7851f5`

[3]`https://git.frama-c.com/pub/pyrat/-/tree/vnncomp2024`

[4]`https://github.com/wu-haoze/Marabou/tree/vnn-comp-23`

[5]`https://github.com/wu-haoze/Marabou/tree/vnn-comp-24`

Table 4: Proportion of "verified" unverifiable instances. It is a measurement of **unsoundness** of verifiers. Any verified instance indicates a bug in the NN verifier. "-" indicates the model is not supported by the verifier. The number of unverifiable instances used in each setting corresponds to "hidden CEX" in Table 2 (only unverifiable instances with hidden counterexamples not found by adversarial attacks are kept, as mentioned in Section 4.2).

| Model | | | Verified (%) | | | | | | |
|---|---|---|---|---|---|---|---|---|---|
| Name | $\epsilon$ | Input size | ABC-A | ABC-I | NSAT-A | NSAT-I | PyRAT | MB-23 | MB-24 |
| CNN AvgPool | 0.2 | $1 \times 5 \times 5$ | 0 | 0 | 0 | 0 | 0 | - | - |
| CNN AvgPool | 0.2 | $3 \times 5 \times 5$ | **100** | **100** | **100** | **100** | 0 | - | - |
| CNN AvgPool | 0.5 | $1 \times 5 \times 5$ | **100** | **66.7** | **100** | **33.3** | 0 | - | - |
| CNN AvgPool | 0.5 | $3 \times 5 \times 5$ | 0 | 0 | 0 | 0 | 0 | - | - |
| CNN Tanh | 0.2 | $1 \times 5 \times 5$ | 0 | 0 | 0 | 0 | 0 | 0 | 0 |
| CNN Tanh | 0.2 | $3 \times 5 \times 5$ | 0 | 0 | 0 | 0 | 0 | **100** | 0 |
| CNN Sigmoid | 0.2 | $1 \times 5 \times 5$ | 0 | 0 | 0 | 0 | 0 | 0 | 0 |
| CNN Sigmoid | 0.2 | $3 \times 5 \times 5$ | 0 | 0 | 0 | 0 | 0 | 0 | 0 |
| ViT | 0.2 | $1 \times 5 \times 5$ | 0 | 0 | 0 | 0 | - | **90** | - |
| ViT | 0.2 | $3 \times 5 \times 5$ | 0 | 0 | 0 | 0 | - | **100** | - |

*Results on other models*: For "CNN 1 Conv", "CNN 2 Conv", "CNN 3 Conv" and the MLP models, no verifier claimed any unverifiable instance as verified. We thus do not create individual rows for these models (their results would have been all 0).

Table 5: Proportion of falsified unverifiable instances. For every model architecture, we average the results on different $\epsilon$ values and input sizes.

| Model Architecture | ABC-A | ABC-I | NSAT-A | NSAT-I | PyRAT | MB-23 | MB-24 |
|---|---|---|---|---|---|---|---|
| CNN 1 Conv | 0 | 0 | 0 | 0 | 0 | 0 | 2.8 |
| CNN 2 Conv | 0 | 0 | 0 | 0 | 0 | 0 | 0 |
| CNN 3 Conv | 2.5 | 3.6 | 0 | 0 | 0 | 0 | 0 |
| CNN AvgPool | 17.5 | 0 | 7.5 | 10.0 | 0 | 25.0 | 25.0 |
| CNN Tanh | 0 | 0 | 0 | 0 | 0 | 0 | 0 |
| CNN Sigmoid | 0 | 65.0 | 0 | 55.0 | 0 | 55.0 | 50.0 |
| ViT | 0 | 0 | 0 | 0 | - | 0 | - |
| MLP 4 Hidden | 5.5 | 0 | 11.1 | 11.1 | 0 | 0 | 0 |
| MLP 5 Hidden | 12.5 | 25.0 | 12.5 | 25.0 | 0 | 0 | 0 |

3. **Our benchmark includes many challenging regular instances.** As shown in Table 6, results vary significantly across verifiers and models, and in most settings, the proportion of verified instances is not high. This demonstrates that our regular instances also reveal performance differences and limitations among different NN verifiers, as a side benefit. Given that these NN verifiers already exhibit limited performance on our relatively small-scale models and data built with pre-defined counterexamples, we have focused on this scale in this paper, as mentioned in Section 3.5.

**Analysis of discovered bugs.** For the bugs on CNN AvgPool models, triggered for $\alpha,\beta$-CROWN and NeuralSAT, we find that they are rooted in the auto_LiRPA package (Xu et al., 2020) leveraged by both $\alpha,\beta$-CROWN and NeuralSAT in their bound computation. Specifically, we find that auto_LiRPA previously assumed that "strides" and "kernel size" are equal in AvgPool layers and silently produced unsound results when their assumption does not hold in our models. Counterexamples provided by SoundnessBench successfully led to the discovery of the bug in auto_LiRPA. We shared our test cases that helped the developers of auto_LiRPA fix the bug. The fix has been reflected in the auto_LiRPA package and adopted by both $\alpha,\beta$-CROWN and NeuralSAT in their latest version. For the bugs of Marabou on Tanh and ViT, the authors of Marabou conjectured that there might be bugs in the interaction with the quadratic programming engine in Gurobi, but the specific cause remains unknown.

Table 6: Proportion of verified regular instances. For every model architecture, we average the results on different $\epsilon$ values and input sizes.

| Model Architecture | ABC-A | ABC-I | NSAT-A | NSAT-I | PyRAT | MB-23 | MB-24 |
|---|---|---|---|---|---|---|---|
| CNN 1 Conv | 45.0 | 35.0 | 47.5 | 35.0 | 25.0 | 7.5 | 45.0 |
| CNN 2 Conv | 32.5 | 27.5 | 32.5 | 27.5 | 12.5 | 5.0 | 22.5 |
| CNN 3 Conv | 17.5 | 20.0 | 20.0 | 20.0 | 12.5 | 0 | 20.0 |
| CNN AvgPool | 75.0$^\dagger$ | 70.0$^\dagger$ | 75.0$^\dagger$ | 70.0$^\dagger$ | 0 | - | - |
| CNN Tanh | 0 | 0 | 0 | 0 | 0 | 50.0$^\dagger$ | 0 |
| CNN Sigmoid | 10.0 | 10.0 | 10.0 | 10.0 | 10.0 | 10.0 | 5.0 |
| ViT | 0 | 0 | 0 | 0 | - | 100.0$^\dagger$ | - |
| MLP 4 Hidden | 55.0 | 75.0 | 55.0 | 85.0 | 45.0 | 0 | 45.0 |
| MLP 5 Hidden | 45.0 | 65.0 | 50.0 | 75.0 | 15.0 | 20.0 | 15.0 |

$^\dagger$Denotes verification results that are likely unsound given bugs identified on unverifiable instances in Table 4.

Table 7: Average number of hidden counterexamples (up to 10) for the ablation study, evaluated across 5 different random seeds.

| Training Settings | | | Margin objective | | Perturbation window size | | | |
|---|---|---|---|---|---|---|---|---|
| Model | $\epsilon$ | Input size | Disabled | Enabled | 1 | 10 | 100 | 300 |
| CNN 1 Conv | 0.2 | $1 \times 5 \times 5$ | 1.2 | **6.2** | 0 | 0 | 4.0 | **6.2** |
| CNN 1 Conv | 0.2 | $3 \times 5 \times 5$ | 1.2 | **10.0** | 0 | 0 | 8.0 | **10.0** |
| CNN 1 Conv | 0.5 | $1 \times 5 \times 5$ | 0.4 | **7.4** | 0 | 0 | 3.6 | **7.4** |
| CNN 1 Conv | 0.5 | $3 \times 5 \times 5$ | 0 | **5.8** | 0 | 0 | **6.6** | 5.8 |

## 4.4 Ablation study

In this section, we conduct an ablation study evaluating the individual contributions of the two techniques proposed in Section 3.4, *margin objective* and *perturbation sliding window*. For this study, we use the CNN 1 Conv architecture with perturbation radii $\epsilon \in \{0.2, 0.5\}$ and input sizes $1 \times 5 \times 5$ and $3 \times 5 \times 5$. For each experiment, we vary the setting of the ablation subject (i.e., whether the margin objective is enabled and the maximum window size, respectively), and we train 5 models with different random seeds for each setting. We evaluate the average number of hidden counterexamples produced in each setting.

Table 7 shows the results. Models trained with the margin objective consistently outperform their counterparts trained without the margin objective, demonstrating the effectiveness of the margin objective in producing many more hidden counterexamples. When the window size is set to 1, the training is equivalent to the original adversarial training, and no hidden counterexample can be produced. The best results are achieved by setting the window size to 100 in one case and to 300 in the other three cases. This demonstrates the effectiveness of our perturbation sliding window.

## 5 Conclusion and Discussion

We present SoundnessBench, a benchmark with hidden counterexamples for testing the soundness of NN verifiers. Through a two-objective training framework with two training techniques, margin objective and perturbation sliding window, we have created a benchmark incorporating 26 models across nine distinct NN architectures, 206 unverifiable instances with hidden counterexamples, and 260 regular instances. Notably, SoundnessBench revealed bugs in three well-established NN verifiers: $\alpha,\beta$-CROWN, NeuralSAT, and Marabou 2023. Our benchmark has helped the authors of auto_LiRPA, which is used in both $\alpha,\beta$-CROWN and NeuralSAT, fix the bug we identified. We believe SoundnessBench and our proposed methodology can serve as a valuable resource for developers building more reliable NN verifiers and more trustworthy artificial intelligence, thus broadly benefiting safety-critical domains that utilize NN verifiers in the future.

**Limitations.** This work has several limitations. First, our models and data are relatively small-scale, as discussed in Section 3.5, but they remain useful for testing the soundness of NN verifiers as our results demonstrate. Second, we mainly use PGD and AutoAttack, which are among the strongest and most commonly used attacks both in verifiers (such as $\alpha,\beta$-CROWN) and in the literature. While PGD and AutoAttack cannot cover new attacks that may appear in the future, it is only feasible for us to adopt currently existing attacks, and future work may consider new attacks as we discuss below.

**Benchmark growth and maintenance.** Our benchmark can be extended to consider new settings and new attacks that may emerge. We suggest closely monitoring the development of various verifiers and performance benchmarks to maintain our benchmark: (1) Results should be updated as new releases of verifiers appear; (2) If a new attack is introduced in a verifier, we need to test the resilience of our currently hidden counterexamples against such attacks. If our counterexamples are no longer hidden, we should adapt our training to incorporate such stronger attacks by replacing the PGD attack with the new attack algorithm, as the attack component in our training pipeline is independent of the other components; (3) If a new model architecture or data setting becomes popular in the NN verification community and is supported by multiple verifiers, we can extend our benchmark to support such settings. We welcome contributions from the community to help maintain and expand SoundnessBench.

**Increasing the soundness and reliability of NN verifiers.** Beyond the development of novel and advanced NN verification algorithms, the correct implementation of algorithms is critical to maintaining the soundness of verifiers, which can be challenging to validate systematically. Most NN verification papers and VNN-COMP primarily focus on the performance of NN verifiers, while the reliability of NN verifiers is currently understudied. We thus encourage the community to dedicate more effort to testing the soundness of NN verifiers. Specifically, we recommend that researchers: (1) stress-test new NN verifiers with soundness benchmarks and tools when new algorithms or implementation modifications are introduced, and develop new soundness benchmarks simultaneously as new settings are studied; (2) incorporate measures and results of soundness testing into research papers and technical reports; and (3) document the assumptions of algorithms and implementations and properly handle unsupported cases. By prioritizing both performance and soundness, the community can build more capable and reliable NN verifiers and verified NN-based models for various safety-critical applications.

### Acknowledgments

This project is supported in part by NSF 2048280, 2331966 and ONR N00014-23-1-2300:P00001. Huan Zhang is supported in part by the AI2050 program at Schmidt Sciences (AI2050 Early Career Fellowship) and NSF (IIS-2331967).

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

## A   Illustration for the Training Algorithm

In Algorithm 1, we illustrate our algorithm for training models with hidden counterexamples, as proposed in Section 3.4.

---

**Algorithm 1** Training Models with Hidden Counterexamples

---

**Inputs:** Dataset $\mathcal{D} = \{(\boldsymbol{x}_0^{(i)}, y^{(i)})\}_{i=1}^{2n}$; initialized model $\boldsymbol{f_\theta}$; attack method $\mathcal{A}$ (i.e., PGD); margin cap $\lambda$; perturbation sliding window size $w$; epochs $T$; optimizer $\mathtt{Opt}$.

**Output:** Trained model $\boldsymbol{f_\theta}$

1:   *// Initialize perturbation sliding windows*
2:   **for** $i = 1, \ldots, 2n$ **do**
3:      $\mathbb{W}^{(i)} \leftarrow \emptyset$
4:   **end for**
5:   *// Main training loop*
6:   **for** $t = 1, 2, \ldots, T$ **do**
7:      *// Generate new adversarial perturbations and update perturbation sliding windows*
8:      **for** $i = 1, \ldots, 2n$ **do**
9:         $\boldsymbol{\delta}^{(i)} \leftarrow \mathcal{A}(\boldsymbol{f_\theta}, \boldsymbol{x}_0^{(i)}, y^{(i)}, \epsilon)$
10:       $\mathbb{W}^{(i)} \leftarrow (\mathbb{W}^{(i)} \cup \{\boldsymbol{\delta}^{(i)}\})$ and truncate to keep the most recent $w$ elements
11:     **end for**
12:     *// Margin objective for pre-defined counterexamples (Eq. (10))*
13:     $\mathcal{L}_{\text{CEX}} = \dfrac{1}{n} \sum_{i=1}^{n} \max\{0, \boldsymbol{f}_{y^{(\text{unv},i)}}(\boldsymbol{x}^{(\text{unv},i)}) - \boldsymbol{f}_{y_{\text{CEX}}^{(i)}}(\boldsymbol{x}_{\text{CEX}}^{(i)}) + \lambda)\}$
14:     *// Adversarial training with perturbation sliding windows (Eq. (11))*
15:     $\mathcal{L}_{\text{adv}} = \dfrac{1}{2n} \sum_{1 \leq i \leq 2n} \dfrac{1}{|\mathbb{W}^{(i)}|} \sum_{\boldsymbol{\delta} \in \mathbb{W}^{(i)}} \mathcal{L}_{\text{CE}}(\boldsymbol{f}(\boldsymbol{x}_0^{(i)} + \boldsymbol{\delta}), y^{(i)})$.
16:     *// Update the model $\boldsymbol{f_\theta}$ with a combined objective (Eq. (12))*
17:     $\mathcal{L}(\boldsymbol{\theta}) = \mathcal{L}_{\text{CEX}}(\boldsymbol{\theta}) + \mathcal{L}_{\text{adv}}(\boldsymbol{\theta})$
18:     $\boldsymbol{\theta} \leftarrow \mathtt{Opt.step}(\boldsymbol{\theta}, \nabla_{\boldsymbol{\theta}} \mathcal{L}(\boldsymbol{\theta}))$
19:   **end for**
20:   **return** Trained model $\boldsymbol{f_\theta}$

---

## B   Comparison between Benchmarks

In Table 8, we compare our benchmark with existing ones compiled in VNN-COMP 2024 (Brix et al., 2024). Our benchmark serves a unique purpose for testing the soundness of NN verifiers while others all focus on performance. Meanwhile, our benchmark covers diverse model architectures, while most existing benchmarks focus on one or two model architectures. Additionally, while we focus on relatively small models and data as discussed in Section 3.5, our scale is still larger than many of the existing benchmarks.

## C   Synthetic Bugs

To further test our soundness benchmark, we introduce synthetic bugs into $\alpha,\beta$-CROWN and NeuralSAT using *shrunk input & intermediate bounds* and *randomly dropped domains*:

1. **Shrunk input & intermediate bounds**: We modify the verifiers and deliberately shrink input and intermediate verified bounds in an unsound way, by a factor denoted as $\alpha$ ($0 < \alpha \leq 1$). The input bounds correspond to the perturbation radius $\epsilon$, and we reduce $\epsilon$ into $(1-\alpha) \cdot \epsilon$. Intermediate bounds are output bounds of intermediate layers as required by the verifiers to perform linear relaxation in LiRPA. Suppose $[\underline{h}_k, \bar{h}_k]$ denotes the lower and upper bound, respectively, of an intermediate layer $k$, and $h_k(\boldsymbol{x}_0)$ is the output of layer $k$ when an unperturbed input is given. We then shrink the

Table 8: Comparison between different benchmarks. Except for our benchmark, others are cited from VNN-COMP 2024 (Brix et al., 2024).

| Benchmark | Purpose | Model architecture | # Params | Input size |
|---|---|---|---|---|
| Acas XU | | MLP | 13k | 5 |
| CCTSDB | | Complex | 100k | 2 |
| cGAN | | CNN, ViT | 500k - 68M | 5 |
| cifar100 | | ResNet | 2.5M - 3.8M | 3072 |
| Collins Aerospace | | Complex | 1.8M | 1.2M |
| Collins RUL CNN | | CNN | 60k - 262k | 400 - 800 |
| CORA | | MLP | 575k, 1.1M | 784, 3072 |
| Dist Shift | | MLP | 342k - 855k | 792 |
| LinearizeNN | Performance | MLP | 203k | 4 |
| LSNC | | Complex | 210k | 8 |
| Metaroom | | Conv | 466k - 7.4M | 5376 |
| ml4acopf | | Complex | 4k - 680k | 22 - 402 |
| NN4Sys | | MLP | 33k - 37M | 1-308 |
| safeNLP | | MLP | 4k | 30 |
| tinyimagenet | | ResNet | 3.6M | 9408 |
| TLL Verify Bench | | Two-Level Lattice NN | 17k - 67M | 2 |
| Traffic Signs Recognition | | CNN | 905k - 1.7M | 2.7k - 12k |
| VGGNet16 | | CNN | 138M | 150k |
| ViT | | ViT | 68k - 76k | 3072 |
| Yolo | | ResNet | 22k - 37M | 1 - 308 |
| SoundnessBench (ours) | **Soundness** | MLP, CNN (ReLU, Sigmoid, Tanh, AvgPool), ViT | 5k - 3M | 25 - 75 |

intermediate bounds by pushing them towards $h_k(\boldsymbol{x}_0)$:

$$[\underline{h}_k, \bar{h}_k] \rightarrow \left[\underline{h}_k + \alpha\left(h_k(\boldsymbol{x}_0) - \underline{h}_k\right), \overline{h}_k + \alpha\left(h_k(\boldsymbol{x}_0) - \overline{h}_k\right)\right] \quad \text{for } 0 < \alpha \leq 1.$$

Shrinking the input and intermediate bounds breaks the soundness of verification, which is thus used as a synthetic bug.

2. **Randomly dropped domains**: Branch-and-bound used in NN verifiers iteratively branches (or splits) the verification problem into smaller subproblems, where each subproblem involves a domain of input bounds or intermediate bounds, and tighter verified bounds can be computed for the smaller domains. In this synthetic bug, we randomly drop a proportion of new domains created in each branch-and-bound iteration, and thereby break the soundness of the verifier. We use $\beta$ ($0 < \beta \leq 1$) to denote the ratio of randomly dropped domains.

**Findings** In the experiment, we report the minimum $\alpha$ or $\beta$ values that make at least one unverifiable instance to be claimed as verified by the modified verifier with the synthetic bug, as well as the minimum $\alpha$ or $\beta$ values that make all unverifiable instances claimed as verified. We use a binary search to find the minimum $\alpha$ and $\beta$ values. Results are shown in Tables 9 and 10. The results show that when a sufficiently

Table 9: Verification results on unverifiable instances for verifiers with synthetic bugs using shrunk input & intermediate bounds. $\alpha$ denotes the factor of shrunk bounds (a larger $\alpha$ value implies a more significant synthetic bug).

| Model | | | Min $\alpha$ for One "Verified" | | Min $\alpha$ for All "Verified" | |
|---|---|---|---|---|---|---|
| Name | $\epsilon$ | Input size | ABC-I | NSAT-I | ABC-I | NSAT-I |
| CNN 1 Conv | 0.2 | $1 \times 5 \times 5$ | 0.28 | 0.27 | 0.62 | 0.33 |
| CNN 1 Conv | 0.2 | $3 \times 5 \times 5$ | 0.50 | 0.48 | 0.75 | 0.62 |
| CNN 1 Conv | 0.5 | $1 \times 5 \times 5$ | 0.27 | 0.26 | 0.55 | 0.31 |
| CNN 1 Conv | 0.5 | $3 \times 5 \times 5$ | 0.36 | 0.35 | 0.39 | 0.38 |

Table 10: Verification results on unverifiable instances for verifiers with synthetic bugs using randomly dropped domains. $\beta$ denotes the ratio of randomly dropped domains.

| Model | | | Min $\beta$ for One "Verified" | | Min $\beta$ for All "Verified" | |
|---|---|---|---|---|---|---|
| Name | $\epsilon$ | Input size | ABC-I | NSAT-I | ABC-I | NSAT-I |
| CNN 1 Conv | 0.2 | $1 \times 5 \times 5$ | 0.39 | 0.40 | 0.43 | 0.42 |
| CNN 1 Conv | 0.2 | $3 \times 5 \times 5$ | 0.44 | 0.44 | 0.44 | 0.46 |
| CNN 1 Conv | 0.5 | $1 \times 5 \times 5$ | 0.38 | 0.39 | 0.40 | 0.41 |
| CNN 1 Conv | 0.5 | $3 \times 5 \times 5$ | 0.45 | 0.44 | 0.46 | 0.44 |

large $\alpha$ or $\beta$ value is applied ($0.26 \sim 0.50$ for $\alpha$, and $0.38 \sim 0.45$ for $\beta$), our soundness benchmark can reveal the synthetic bug, as at least one unverifiable instance is claimed as verified by the modified verifier. If $\alpha$ or $\beta$ values are further increased up to 0.75 for $\alpha$ and 0.46 for $\beta$, all the unverifiable instances are claimed as verified. These results demonstrate the effectiveness of our soundness benchmark when synthetic bugs are introduced into verifiers.

## D    Multiple Random Seeds

To evaluate the stability of our results, we conduct experiments with multiple random seeds for both training and verification, and find that our results overall remain stable across different seeds.

Table 11: Number of hidden counterexamples (out of 10 unverifiable instances) across different training seeds.

| Model | Run 1 | Run 2 | Run 3 | Run 4 | Run 5 |
|---|---|---|---|---|---|
| CNN AvgPool ($\epsilon = 0.5$ and input size $1 \times 5 \times 5$) | 9 | 10 | 9 | 9 | 4 |
| ViT ($\epsilon = 0.2$ and input size $3 \times 5 \times 5$) | 10 | 10 | 8 | 9 | 8 |

**Training with Multiple Seeds**   We select two representative models that consistently expose verifier bugs: (1) CNN AvgPool ($\epsilon = 0.5$ and input size $1 \times 5 \times 5$) and (2) ViT ($\epsilon = 0.2$ and input size $3 \times 5 \times 5$). For each setting, we run training four additional times using different random seeds. Combined with results in Section 4.2, we now have a total of five runs for each of these models. Results are shown in Table 11. The number of hidden counterexamples remains consistently high across different runs. The CNN AvgPool model shows slight variation, but still produces 9 or more hidden counterexamples in four of the five runs. The ViT model demonstrates even stronger consistency, producing 8 or more hidden counterexamples in all five runs. These results indicate that our approach is largely stable to randomness in training.

Table 12: Proportion of falsified unverifiable instances on MNIST.

| Model Architecture | Hidden CEX | ABC-A | ABC-I | NSAT-A | NSAT-I | PyRAT | MB-23 | MB-24 |
|---|---|---|---|---|---|---|---|---|
| CNN 1 Conv | 8 | 0 | 0 | 0 | 0 | 0 | 0 | 0 |
| CNN 2 Conv | 9 | 33.3 | 33.3 | 0 | 11.1 | 0 | 0 | 0 |
| CNN AvgPool | 6 | 0 | 0 | 0 | 0 | 0 | 0 | 0 |
| CNN Tanh | 9 | 0 | 0 | 0 | 0 | 0 | 0 | 0 |
| CNN Sigmoid | 3 | 33.3 | 66.7 | 0 | 66.7 | 0 | 0 | 0 |
| ViT | 3 | 66.7 | 66.7 | 66.7 | 66.7 | - | 0 | - |
| MLP 4 Hidden | 10 | 30.0 | 30.0 | 20.0 | 30.0 | 0 | 0 | 0 |
| MLP 6 Hidden | 9 | 22.2 | 22.2 | 22.2 | 22.2 | 0 | 0 | 0 |

**Verification with Multiple Seeds.** We also conduct an experiment on the verifier $\alpha,\beta$-CROWN , the only verifier that explicitly takes a random seed as an input argument, to examine how randomness affects verification outcomes. Specifically, we run $\alpha,\beta$-CROWN with different random seeds on two CNN AvgPool models: (1) $\epsilon = 0.2$ and input size $1 \times 5 \times 5$ and (2) $\epsilon = 0.2$ and input size $3 \times 5 \times 5$. For the first model, results show that the verification outcomes remain consistent across different random seeds for all instances. For unverifiable instances, ABC-A consistently verifies 100% of instances across all five random seeds, while ABC-I consistently verifies 75% and times out on 25%. For regular instances, both ABC-A and ABC-I consistently verify 100% of instances across all seeds. This shows that the soundness bug in $\alpha,\beta$-CROWN can be reliably triggered regardless of the random seed for verification. For the second model, which contains several unverifiable instances $\alpha,\beta$-CROWN successfully falsifies, we observe that the proportion of falsified instances varies slightly with random seeds (ranging from 55.6% to 66.7% falsified across five seeds for both ABC-A and ABC-I), while results on regular instances remain stable (100% timeout for both ABC-A and ABC-I across all seeds). This behavior is due to the PGD attack used in $\alpha,\beta$-CROWN for falsification, which uses random initialization. Overall, our results still remain largely stable.

## E    Results on MNIST Dataset

In addition to our main experiments on synthetic data, we conduct additional experiments on MNIST (LeCun et al., 2010), where the inputs are real images with sizes $1 \times 28 \times 28$ much larger than those of our synthetic data. We train new models with various architectures and run all the verifiers on models they support. Results are shown in Tables 12 to 15. Our training pipeline successfully produces MNIST models with hidden counterexamples, as shown in Tables 12 and 13. For example, for model architectures where bugs have previously been identified on synthetic data, there are 6, 9 and 3 hidden counterexamples for models CNN AvgPool, CNN Tanh and ViT, respectively. Unsound verification and verifier bugs are still found on some of the new unverifiable instances on MNIST. Specifically, Marabou 2023 still wrongly verifies all the unverifiable instances on ViT, as shown in Table 13. However, on these relatively larger models, NN verifiers with bugs are less likely to produce unsound results and expose bugs, as no verifier verifies any instance besides Marabou 2023 on ViT. These MNIST models are not able to expose the bugs of $\alpha,\beta$-CROWN and NeuralSAT on CNN AvgPool or Marabou 2023 on CNN Tanh. Therefore, we mainly focus on relatively small settings for our purpose of testing the soundness rather than evaluating the performance of NN verifiers, as discussed in Section 3.5.

Table 13: Proportion of "verified" unverifiable instances on MNIST.

| Model Architecture | Hidden CEX | ABC-A | ABC-I | NSAT-A | NSAT-I | PyRAT | MB-23 | MB-24 |
|---|---|---|---|---|---|---|---|---|
| CNN 1 Conv | 8 | 0 | 0 | 0 | 0 | 0 | 0 | 0 |
| CNN 2 Conv | 9 | 0 | 0 | 0 | 0 | 0 | 0 | 0 |
| CNN AvgPool | 6 | 0 | 0 | 0 | 0 | 0 | - | - |
| CNN Tanh | 9 | 0 | 0 | 0 | 0 | 0 | 0 | 0 |
| CNN Sigmoid | 3 | 0 | 0 | 0 | 0 | 0 | 0 | 0 |
| ViT | 3 | 0 | 0 | 0 | 0 | - | **100** | - |
| MLP 4 Hidden | 10 | 0 | 0 | 0 | 0 | 0 | 0 | 0 |
| MLP 6 Hidden | 9 | 0 | 0 | 0 | 0 | 0 | 0 | 0 |

Table 14: Proportion of falsified regular instances on MNIST. The number of regular instances for every model architecture is the same as the number of unverifiable instances.

| Model Architecture | ABC-A | ABC-I | NSAT-A | NSAT-I | PyRAT | MB-23 | MB-24 |
|---|---|---|---|---|---|---|---|
| CNN 1 Conv | 0 | 0 | 0 | 0 | 0 | 0 | 0 |
| CNN 2 Conv | 0 | 0 | 0 | 0 | 0 | 0 | 0 |
| CNN AvgPool | 0 | 16.7 | 16.7 | 0 | 0 | 0 | 0 |
| CNN Tanh | 0 | 0 | 0 | 0 | 11.1 | 0 | 0 |
| CNN Sigmoid | 0 | 33.3 | 33.3 | 0 | 33.3 | 0 | 0 |
| ViT | 0 | 0 | 0 | 0 | - | 0 | - |
| MLP 4 Hidden | 0 | 0 | 0 | 0 | 0 | 0 | 0 |
| MLP 6 Hidden | 22.2 | 33.3 | 22.2 | 33.3 | 0 | 0 | 0 |

Table 15: Proportion of verified regular instances on MNIST. The number of regular instances for every model architecture is the same as the number of unverifiable instances.

| Model Architecture | ABC-A | ABC-I | NSAT-A | NSAT-I | PyRAT | MB-23 | MB-24 |
|---|---|---|---|---|---|---|---|
| CNN 1 Conv | 0 | 0 | 0 | 0 | 0 | 0 | 0 |
| CNN 2 Conv | 0 | 0 | 0 | 0 | 0 | 0 | 0 |
| CNN AvgPool | 0 | 0 | 0 | 0 | 0 | - | - |
| CNN Tanh | 0 | 0 | 0 | 0 | 0 | 0 | 0 |
| CNN Sigmoid | 0 | 0 | 0 | 0 | 0 | 0 | 0 |
| ViT | 0 | 0 | 0 | 0 | - | $100^{\dagger}$ | - |
| MLP 4 Hidden | 0 | 0 | 0 | 0 | 0 | 0 | 0 |
| MLP 6 Hidden | 0 | 0 | 0 | 0 | 0 | 0 | 0 |

$^{\dagger}$Denotes verification results that are likely unsound given bugs identified on unverifiable instances in Table 13.

