# OpenReview forum: "SoundnessBench: A Soundness Benchmark for Neural Network Verifiers"
_TMLR — Accepted by TMLR_

### Review · Reviewer_mpf6 · 2025-10-27

**Summary Of Contributions:**

Summary of Contributions:
This work mainly proposes a soundness benchmark named SoundnessBench to evaluate ability of different neural network verifiers. To make the verification task more challenging, SoundnessBench introduces a model training strategy with two objectives to make counterexamples more concealed. The benchmarking evaluation demonstrate that SoundnessBench is able to provide counterexamples that cannot be identified by state-of-art neural network verifiers.

Strength:
+ The benchmark provides ground truth for hard examples in the evaluation of neural network verifiers. The proposed training framework is able to generate counterexamples that are resistant to strong adversarial attacks.
+ The benchmark evaluation finds bugs in state-of-art neural network verifier methods.

Weakness:
+ The scale and scope of data is limited. Whether counterexamples in small scale data can reflect real-world neural network verification scenarios is unknown. It is necessary to add experiment with larger scale data, as other benchmarks and neural network verification methods include these parts of experiments.
+ Although the benchmark is on “soundness”, but the clear definition of “soundness” is missing. It is better to provide a definition of “soundness” so that we can compare two NN verifier methods on this evaluation metric.

**Audience:**

Yes

**Audience Explanation:**

The audiences that work on neural network verifiers or trustworthy AI topics will be interested in the findings of this paper.

**Broader Impact Concerns:**

There are not concerns that that would require adding a Broader Impact Statement.

**Claims And Evidence:**

Yes

**Claims Explanation:**

Generally, the claims and contributions in the paper is supported by accurate, convincing and clear evidence.
The paper's central claim is supported by empirical evidence through the successful discovery of real, actionable bugs in three state-of-the-art neural network verifiers.

**Requested Changes:**

Critical:
+ Benchmarking evaluation with larger scale and scope of data that can better reflect real-world neural network verification scenarios.
+ A clear definition on the key word “soundness” and benchmarking evaluation that could reflect that key word.

Strengthen the work:
+ A table to compare SoundnessBench and existing benchmark works on NN verifiers.
+ An algorithm to make the training strategy in Section 3.4 clearer.
+ Handle duplicate parts on problem definition in Section 2.1 and Section 3.1.
+ Provide prospective on future research direction on NN verifiers based on benchmarking evaluation findings.

---

> ### Author Response · Authors · 2025-11-17
>
> We thank the reviewer for the positive review, and we appreciate the constructive and detailed feedback for us to extend the data scale and scope of our benchmark, improve the presentation and extend discussions. We respond to the requested changes and additional comments below:
>
> ## Scale and scope
>
> > Benchmarking evaluation with larger scale and scope of data that can better reflect real-world neural network verification scenarios.
>
> We conducted additional experiments on the MNIST dataset, where the inputs are real images with sizes much larger than those of our synthetic data. We trained new models with various architectures, including MLP, CNN including those with AvgPool, Sigmoid, or Tanh, and ViT. We ran all the verifiers on models they support.
>
> Complete results can be found in Appendix E. Here we show partial results of the proportion of “verified” unverifiable instances, on models and verifiers where bugs were previously found on our synthetic data.
>
> | Model    | $\epsilon$ | Input size | Hidden CEX | ABC-A | ABC-I | NSAT-A | NSAT-I | MB-23 |
> |-|-|-|-|-|-|-|-|-|
> | CNN AvgPool  | 0.4    | 1×28×28 | 6 | **0%** | **0%** | **0%** | **0%** | 0% |
> | CNN Tanh  | 0.4    | 1×28×28 | 9 | 0% | 0% | 0% | 0% | **0%** |
> | ViT  | 0.4    | 1×28×28 | 3 | 0% | 0% | 0% | 0% | **100%** |
>
> We summarize the main findings here:
> 1. Our training pipeline successfully produces MNIST models with hidden counterexamples. For example, for model architectures where bugs have previously been identified on synthetic data, there are 6, 9 and 3 hidden counterexamples for models CNN AvgPool, CNN Tanh and ViT, respectively.
> 2. Unsound verification and verifier bugs are still found on some of the new unverifiable instances on MNIST. Specifically, Marabou 2023 wrongly verifies 100% of unverifiable instances on ViT.
> 3. However, on these relatively larger models, NN verifiers with bugs are less likely to produce unsound results and expose bugs, as no verifier verifies any instance besides Marabou 2023 on ViT. These MNIST models are not able to expose the bugs of α,β-CROWN and NeuralSAT on CNN AvgPool or Marabou 2023 on CNN Tanh.
>
> The last observation is consistent with our motivation of focusing on relatively small scale settings, as we have discussed in the “Scale” paragraph in Section 3.5. In particular, verifiers with bugs are less likely to produce wrong results on larger-scale settings due to the increased difficulty in verification. Verifiers typically require relaxation, and for relatively larger settings, looser relaxation of other correct layers can compensate for the wrong bound computation, making the final verification more conservative. Additionally, although NN verifiers can often verify larger models in other benchmarks, they depend on models being specifically trained, and our models trained with hidden counterexamples are more challenging to verify. Therefore, for our purpose of testing soundness and identifying bugs, it is more effective to use relatively small settings which do not necessarily need to be as realistic as “real-world scenarios”. We may draw an analogy to fuzzing techniques in software engineering, where simple and synthetic but specifically constructed inputs, rather than complicated real-world inputs, are used for testing software and identifying bugs. We have revised our paper and expanded the discussion accordingly in the “Scale” paragraph in Section 3.5.
>
> ## Definition for soundness
>
> > A clear definition on the key word “soundness” and benchmarking evaluation that could reflect that key word.
>
> We have revised the paper to (1) provide explicit definitions of “sound” and “unsound” in Section 2 at the beginning of the “Soundness of NN verifiers” paragraph; and (2) highlight results in Table 4 that give the key measurement of “unsoundness” of verifiers, evaluated as the proportion of “verified” unverifiable instances where counterexamples are known to exist. Any nonzero result indicates an unsound verification claim and a bug in the verifier.

---

> > ### Author Response · Authors · 2025-11-17
> >
> > ## Additional comments
> >
> > > A table to compare SoundnessBench and existing benchmark works on NN verifiers.
> >
> > We have added a table in Appendix B to compare our benchmark with existing ones compiled in VNN-COMP 2024 (Brix et al., 2024). The table covers information such as purpose, model architecture, model size, and input size. Most importantly, our benchmark is unique and serves a completely new purpose -- testing the soundness of NN verifiers, while existing benchmarks commonly focus on evaluating the performance of NN verifiers. Meanwhile, our benchmark covers diverse model architectures, while most existing benchmarks focus on one or two model architectures. Additionally, while we focus on relatively small models and data as explained in Section 3.5 and above in “Scale and scope”, our scale is still larger than many of the existing benchmarks (e.g., six of the existing benchmarks have input sizes under 10).
> >
> > > An algorithm to make the training strategy in Section 3.4 clearer.
> >
> > We have added an algorithm illustration Algorithm 1 in Appendix A to better show our training pipeline.
> >
> > > Handle duplicate parts on problem definition in Section 2.1 and Section 3.1.
> >
> > We have revised Section 2.1 and Section 3.1, removing duplicate parts and only defining the problem for the K-way classification in Section 3.1
> >
> > > Provide prospective on future research direction on NN verifiers based on benchmarking evaluation findings.
> >
> > In addition to the “Benchmark growth and maintenance” paragraph we had in Section 5, we added a new paragraph titled “Increasing the soundness and reliability of NN verifiers” in Section 5. We discussed:
> >
> > Beyond the development of novel and advanced NN verification algorithms, the correct implementation of algorithms is critical to maintaining the soundness of verifiers, which can be challenging to validate systematically. Most NN verification papers and VNN-COMP primarily focus on the performance of NN verifiers, while the reliability of NN verifiers is currently understudied. We thus encourage the community to dedicate more effort to testing the soundness of NN verifiers. Specifically, we recommend that researchers: (1) stress-test new NN verifiers with soundness benchmarks and tools when new algorithms or implementation modifications are introduced, and develop new soundness benchmarks simultaneously as new settings are studied; (2) incorporate measures and results of soundness testing into research papers and technical reports; and (3) document the assumptions of algorithms and implementations and properly handle unsupported cases. By prioritizing both performance and soundness, the community can build more capable and reliable NN verifiers and verified NN-based models for various safety-critical applications.

---

> > > ### Comment · Reviewer_mpf6 · 2025-11-21
> > >
> > > I appreciate the answer of the authors. The explanations address my concerns, and I have no further questions.

---

### Review · Reviewer_jnVN · 2025-10-28

**Summary Of Contributions:**

This work introduces SoundnessBench, a benchmark for evaluating and debugging the soundness of Neural Network (NN) verification methods. Authors train small models in a synthetic task, with the objective to be robust, but missclassify predefined randomly-sampled perturbations. These are known as hidden counter examples. After training, they keep instances where 1) clean samples are correctly classified 2) hidden counter examples are missclassified 3) strong attacks (PGD and Autoattack) do not succeed in finding a counterexample. Then, if a verification methods verifies an instance where a hidden counter examples is known to exist, the instance is marked as a bug. Through this mechanism, authors discover bugs in three leading verification methods. This lead to a fix in the auto_LIRPA library, which affected two of the verification methods.

**Additional Comments:**

Have you tested $\alpha$-$\beta$-CROWN and NeuralSAT on SoundnessBench after the fix in auto_LIRPA? Did the bugs disappear?

**Audience:**

Yes

**Audience Explanation:**

While training models with known adversarial examples is not new [1] and mechanisms to stress-test robustness evaluations are also known [2], the evaluation of such techniques for detecting “bugs” in NN verification methods is novel and interesting for the community.

This benchmark is useful and interesting for TMLR’s audience as it provides mechanisms to stress-test NN verification methods. I expect authors to make their models and instances public to allow others to test their verification methods.

**References**

[1] BadNets: Identifying Vulnerabilities in the Machine Learning Model Supply Chain, IEEE Access, 2019

[2] Increasing confidence in adversarial robustness evaluations, NeurIPS 2022

**Broader Impact Concerns:**

None.

**Claims And Evidence:**

Yes

**Claims Explanation:**

Authors effectively design verification instances where three existing verification methods ($\alpha$-$\beta$-CROWN, Marabou and NeuralSAT) are not sound.

In Sections 3 and 4, authors describe in detail their methodology to construct the verification instances in SoundnessBench.

In Section 4.4, authors ablate the need of two techniques employed in the training of their instances: 1) The perturbation window size and 2) The use of a margin objective. This ablation study shows the need of both techniques to produce a high quantity of instances where verification methods are not sound.

**Requested Changes:**

- **Fix notation:**  In the second line of Section 2, input vectors “x” are represented with lower-case non-bold letters, but then, from the next paragraph onwards, lower-case bold letters are used for vectors and lower-case non-bold for single numbers. Similarly, functions are represented as lower-case non-bold letters regardless of their output being a vector (“f”) or a single number (“s” and “h”). The notation should be homogenized. For example, all of the vectors (including outputs of “f”) should be denoted with lower-case bold letters.

- **Vague discussion of related work:** The closest work to SoundnessBench is [2]. However, the discussion of the relationship between both methods is quite vague, just 4 lines at the end of Section 2. There are many similarities, e.g., hidden adversarial examples are sampled at the boundary for both methods ($r=0.98$ for SoundnessBench and $r=1$ for [2]) and some differences, e.g., [2] only train the last layer of the classifier. It would be great to understand why the methodologies should be different if the desired objective is the same. A broather discussion on this topic is needed. Also, there are many methods like [1] designed to introduce backdoors in neural networks, a discussion on those is also needed.

---

> ### Author Response · Authors · 2025-11-17
>
> We thank the reviewer for the positive review and constructive feedback for us to improve the notations and discussion of related work. We respond to the requested changes and additional comments below:
>
> ## Notations
>
> We have fixed the notations for $\boldsymbol{x}$, $\boldsymbol{\theta}$, $\boldsymbol{\delta}$, and $\boldsymbol{f}$ to use lower-case bold letters for all the vectors, including function $\boldsymbol{f}$ with a vector output.
>
> ## Discussion of related work
>
> Regarding comparison with Zimmermann et al. (2022) ([2] cited by the reviewer): In our Section 2, we have originally mentioned the key difference compared to Zimmermann et al. (2022), as “their counterexamples were not designed to be hidden against strong attacks”. **We would like to clarify that while they have pre-defined counterexamples, such counterexamples are not “hidden counterexamples”** (unlike what the reviewer mentioned). Thus, **the desired objective is in fact not the same** -- we not only want pre-defined counterexamples to really become counterexamples, but also require the counterexamples to be hidden counterexamples, i.e., not easily found by falsification mechanisms in NN verifiers. This is a unique and novel objective for our problem of testing the soundness of NN verifiers. Our method is designed to achieve this more challenging objective, while Zimmermann et al. (2022) did not have any consideration to make counterexamples hidden.
>
> We have extended the discussion of related work in Section 2, and we now have a standalone paragraph for “Pre-defined counterexamples” discussing Zimmermann et al. (2022) and backdoor attacks including Gu et al., (2017) ([1] cited by the reviewer). For Zimmermann et al. (2022), we better clarified the difference in the objective and thus a requirement for a new training design. For backdoor attacks, we discussed the similarity (“injected wrong behaviors into NNs”) and also differences, as our pre-defined counterexamples are instance-level for the setting of NN verification tested on individual instances, unlike model-level triggers in backdoor attacks; and we require the model to be mostly robust on the input region that contains the pre-defined counterexample for each instance, which is also different compared to backdoor attacks.
>
> ## Additional responses
>
> >Have you tested --CROWN and NeuralSAT on SoundnessBench after the fix in auto_LIRPA? Did the bugs disappear?
>
> We have tested both α,β-CROWN and NeuralSAT on our benchmark after fixing the unsoundness bug in auto_LiRPA. Now both α,β-CROWN and NeuralSAT give unknown results until timeout instead of wrongly claiming to verify the unverifiable instances.
>
> >I expect authors to make their models and instances public to allow others to test their verification methods.
>
> Our models and data will definitely be made public. Additionally, we will open-source our code for both running evaluation with our current benchmark and training new models.

---

> > ### Comment · Reviewer_jnVN · 2025-11-20
> >
> > Thank you for the extended discussion on backdoor attacks and for answering my questions.
> >
> > You are right, the adversarial examples produced by [2] ([2] as referenced in my original review) are not hidden, thanks for the clarification. However, I think the paper still lacks a comparison of the methodologies and not just the purpose of the methods. As I discussed in my review, some parts of the method are very similar, e.g., sampling the adversarial example near the neighborhood. Also, even if [2] don't train the adversarial examples to be "hidden", they purposely train to have adversarial examples. Even though some choices of your method like the margin loss or the window size help observing more hidden adversarial examples (Table 7), your mechanism for obtaining hidden adversarial examples is simply the filtering step in point 3 at the beginning of Section 4.2. I agree that the purpose of your method is different than [2], but I insist that the reader needs to understand the methodological differences between both methods as they are the closest ones. For example, in Section 3.3 you could mention the differences in the sampling procedure, in Section 3.4 the differences in the objectives...
> >
> > Thank you.

---

> ### Author Response · Authors · 2025-11-23
>
> We thank the reviewer for the response and helpful suggestions on including more methodological comparison with Zimmermann et al., 2022 ([2] cited by the reviewer). We have revised the paper accordingly. Specifically:
>
> 1. In Section 2, we referred to Section 3 for a methodological comparison with Zimmermann et al. (2022).
>
> 2. At the end of Section 3.3 for data generation, after mentioning generating our pre-defined counterexamples near the boundary of the perturbation ball, we added:
> >This is similar to how Zimmermann et al. (2022) controlled the distance and generated perturbed examples near the boundary of the $ \mathcal{B}(\mathbf{x}_0,\epsilon)$ perturbation ball when creating their dataset for planting counterexamples.
>
> 3. In Section 3.4 for training, at the end of the “Objective for the pre-defined counterexamples” paragraph, we added:
> >This is different from Zimmermann et al. (2022) as they only trained a linear classification head to separate the frozen features of perturbed examples inside the perturbation ball and on the boundary of the perturbation ball, thereby making the examples on the boundary become counterexamples. In contrast, we train the whole model, as we additionally have a more challenging objective of making the counterexamples hidden against adversarial attacks (achieved via the adversarial training objective described below), and we achieve
> the objective for the pre-defined counterexamples here with a margin objective.
>
> 4. In Section 3.4 for training, at the end of the “Objective with adversarial training” paragraph, we added:
> >This differs significantly from Zimmermann et al. (2022) which did not have adversarial training to make counterexamples hidden.
>
>
> Finally, regarding your comment "Even though some choices of your method like the margin loss or the window size help observing more hidden adversarial examples (Table 7), your mechanism for obtaining hidden adversarial examples *is simply the filtering step* in point 3 at the beginning of Section 4.2", we do not think this is precise. The filtering step is for the final evaluation and collecting the outcome of the training, while hidden counterexamples are produced by the training algorithm. Without appropriate training, the filtering step itself does not produce hidden counterexamples.
>
>
> We welcome any additional suggestions for further revision and would be pleased to revise our paper accordingly. Thank you.

---

> > ### Comment · Reviewer_jnVN · 2025-11-24
> >
> > Thanks for the revisions, the methodological differences are clear now.

---

### Review · Reviewer_RC6k · 2025-11-10

**Summary Of Contributions:**

This paper proposes a method of training seemingly robust neural networks (safe against standard adversarial attacks) with intentionally placed adversarial attacks so that the soundness of neural network verifiers can be tested.  They train a variety of networks and use their technique to find bugs in other well known verifiers.

**Audience:**

Yes

**Audience Explanation:**

While this method does not ensure the soundness of verifiers, the fact that bugs have been identified is significant.

**Claims And Evidence:**

Yes

**Claims Explanation:**

The main claim of the paper is that attacks can be planted that can not be found easily using PGD and other traditional adversarial attacks, and that knowing these in advance can lead to finding bugs in other verifiers.  No theorems have been presented, but the methodology of planting these seems correct and is backed by bugs acknowledged by the authors of the other verifiers.  Except for minor changes listed below, I find the paper to be quite thorough.

**Requested Changes:**

Small changes and grammar:
1. "The input can be specified with an input region"
2. "Formally, the NN verification problem can be defined as formally verifying"
3. "An NN verifier is sound if the verification process is rigorously provable" I think soundness should be defined here properly (it does not have to do with provability of verification).
4. "If both Eq. (4) and Eq. (5) are achieved" - I am not sure this is theoretically possible.  While I understand the intent is to imply that the approximate solutions to the argmax can be counterexamples, as stated delta_cex would be an approximate solution to 5.  I think this can be made clearer and more precise.

---

> ### Author Response · Authors · 2025-11-17
>
> We thank the reviewer for the positive review and constructive feedback on improving the writing.
>
> For the first two requested changes, we revised the paper accordingly to avoid redundant words.
>
> Regarding the definition for the soundness, we removed the imprecise wording “verification process is rigorously provable”. Now, we think we have a more clear definition:
>
> >An NN verifier is *sound* if every time the verifier claims that Eq. (1) holds, there exists no counterexample $\mathbf{x}\in\mathcal{C}$ that violates the condition $h(\mathbf{x})>0$.
>
> For “both Eq. (4) and Eq. (5) are achieved”, we revised the paper to make it more clear. In the revised version, we first define $\mathcal{A}(\mathbf{x}_0,\epsilon,y,i)$ that denotes the solution returned by an adversarial attack solving the minimization problem (this becomes the new Eq. (5)). Then, we have Eq. (6) to replace the original Eq. (5), where we use $\mathcal{A}(\mathbf{x}_0,\epsilon,y,i)$ instead of $\hat{\delta}$ in our previous version. In this way, we have avoided using “$\approx$” that might be confusing.
>
> If the reviewer has further suggestions for us to make these more precise, we would be very grateful and willing to revise our writing further.

---

> > ### Comment · Reviewer_RC6k · 2025-12-11
> > **Response**
> >
> > Thank you for fixing the definition and improving the clarity of that sentence.  I have no further suggestions.

---

### Decision · Action_Editor_7Ytz · 2025-12-22

**Recommendation:** Accept as is

**Audience:**

Yes

**Audience Explanation:**

Verification to adversarial attacks is an interesting area inside machine learning, so there are individuals that will find this useful.

**Claims And Evidence:**

Yes

**Claims Explanation:**

The paper introduces "SoundnessBench," a benchmark constructed to strictly evaluate neural network verifier soundness by generating synthetic instances with deliberately hidden counterexamples that are designed to evade standard adversarial attacks. Then, the verification method is assessed on whether the counterexamples are reported or not. The evaluation is limited by the use of small-scale data, but it is still interesting for the community.